# Molecular Interplay between Non-Host Resistance, Pathogens and Basal Immunity as a Background for Fatal Yellowing in Oil Palm (*Elaeis guineensis* Jacq.) Plants

**DOI:** 10.3390/ijms241612918

**Published:** 2023-08-18

**Authors:** Cleiton Barroso Bittencourt, Thalliton Luiz Carvalho da Silva, Jorge Cândido Rodrigues Neto, André Pereira Leão, José Antônio de Aquino Ribeiro, Aline de Holanda Nunes Maia, Carlos Antônio Ferreira de Sousa, Betania Ferraz Quirino, Manoel Teixeira Souza Júnior

**Affiliations:** 1Graduate Program of Plant Biotechnology, Federal University of Lavras, Lavras 37203-202, MG, Brazil; cleiton_court@hotmail.com (C.B.B.); thallitons@gmail.com (T.L.C.d.S.); 2The Brazilian Agricultural Research Corporation, Embrapa Agroenergy, Brasília 70770-901, DF, Brazil; jorgecrn@hotmail.com (J.C.R.N.); andre.leao@embrapa.br (A.P.L.); jose.ribeiro@embrapa.br (J.A.d.A.R.); betania.quirino@embrapa.br (B.F.Q.); 3The Brazilian Agricultural Research Corporation, Embrapa Environment, Jaguarina 13918-110, SP, Brazil; aline.maia@embrapa.br; 4The Brazilian Agricultural Research Corporation, Embrapa Mid-North, Teresina 64008-780, PI, Brazil; carlos.antonio@embrapa.br

**Keywords:** bud rot, hypoxia, molecular plant biology, pathogen, palm oil, fatal yellowing, molecular mechanisms, transcriptomics, metabolomics, multi-omics, non-host resistance, immunity

## Abstract

An oil palm (*Elaeis guineensis* Jacq.) bud rod disorder of unknown etiology, named Fatal Yellowing (FY) disease, is regarded as one of the top constraints with respect to the growth of the palm oil industry in Brazil. FY etiology has been a challenge embraced by several research groups in plant pathology throughout the last 50 years in Brazil, with no success in completing Koch’s postulates. Most recently, the hypothesis of having an abiotic stressor as the initial cause of FY has gained ground, and oxygen deficiency (hypoxia) damaging the root system has become a candidate for stress. Here, a comprehensive, large-scale, single- and multi-omics integration analysis of the metabolome and transcriptome profiles on the leaves of oil palm plants contrasting in terms of FY symptomatology—asymptomatic and symptomatic—and collected in two distinct seasons—dry and rainy—is reported. The changes observed in the physicochemical attributes of the soil and the chemical attributes and metabolome profiles of the leaves did not allow the discrimination of plants which were asymptomatic or symptomatic for this disease, not even in the rainy season, when the soil became waterlogged. However, the multi-omics integration analysis of enzymes and metabolites differentially expressed in asymptomatic and/or symptomatic plants in the rainy season compared to the dry season allowed the identification of the metabolic pathways most affected by the changes in the environment, opening an opportunity for additional characterization of the role of hypoxia in FY symptom intensification. Finally, the initial analysis of a set of 56 proteins/genes differentially expressed in symptomatic plants compared to the asymptomatic ones, independent of the season, has presented pieces of evidence suggesting that breaks in the non-host resistance to non-adapted pathogens and the basal immunity to adapted pathogens, caused by the anaerobic conditions experienced by the plants, might be linked to the onset of this disease. This set of genes might offer the opportunity to develop biomarkers for selecting oil palm plants resistant to this disease and to help pave the way to employing strategies to keep the safety barriers raised and strong.

## 1. Introduction

Throughout the last decade, oil palm (*Elaeis guineensis* Jacq.) has been the source of the most consumable vegetal oils on the planet [1]. Palm oil and palm kernel oil increased in consumption from 64 to approximately 85 million metric tons between the 2013/14 and 2022/23 seasons. In the top oil palm production countries, the palm oils yield an average of four metric tons per hectare, far better than soybean, canola, peanuts, and other known oilseed crops in terms of oil yield [2,3]. Indonesia, Malaysia, Nigeria, Thailand, and Colombia are the top five countries in the world in terms of oil palm harvested area [3].

Brazil has about 200,000 hectares harvested with oil palm nowadays, even though there are over seven million hectares of preferential area for oil palm cultivation in the so-called Legal Amazon Area, an area of more than five million square kilometers comprising nine states, and larger than the Amazon biome itself [4]. In 2010, a Sustainable Palm Oil Production Program was launched that aimed to strengthen the palm oil industry in the country. Such program had as guidelines: (a) the protection of the environment, the conservation of biodiversity, and the rational use of natural resources; (b) the respect for the social function of property; (c) the expansion of oil palm cultivation exclusively in areas already occupied by man; (d) encouraging cultivation to recover degraded areas; (e) social inclusion; and (f) the environmental regularization of rural properties [5]. More than a decade after launching that program, the problem of FY study is very real, especially in Brazil.

A closer look at the reasons behind the failure to succeed in this endeavor will reveal a complex and multi-factor scenario where the Fatal Yellowing (FY) disease, a bud rod disorder of unknown etiology, is one of the top constraints affecting the growth of the oil palm industry in the country [6]. FY symptoms initiate with the yellowing of the leaflets at the base of the intermediate leaves and progress to necrosis of the edges, which spreads to the other leaves. Subsequently, necrosis and dryness of the spear leaf occur, which evolves towards the meristem region, causing decay and culminating in the death of the oil palm plant. Generally, symptoms progress to oil palm death in a few months (acute form) to three years (chronic) [7,8].

The etiology of FY has been a challenge embraced by several research groups in plant pathology throughout the last 50 years, with no success in completing Koch’s third postulate—inoculation of a healthy plant with the cultured microorganism must recapitulated the disease [6]. Furthermore, the exponential growth of cases and the undefined pattern of spread of the disease weaken a possible biotic primary cause [9]. Most recently, the hypothesis of having an abiotic stressor as the initial cause of FY has gained ground [6]. Silveira et al. [10] showed surface compaction of the soils in the area where FY occurs. This condition can lead to soil saturation with water where oxygen deficiency (hypoxia) possibly damages the root system. An imbalance of nutrients such as Copper, Iron, Manganese, and Zinc has been suggested as a possible cause [11]. Silveira et al. [10] found that the evolution of FY symptoms was more pronounced when there was a reduction in Boron and Copper in the soil. On the other hand, the application of iron sulfate in the study conducted by Viégas et al. [11] ruled out Fe deficiency being involved. Muniz [12] reaffirms the same, and also observed that poor aeration reduces the redox potential of the soil, increasing the concentration of reduced ions, such as Fe^3+^, NO^3+^; Mn^3+^, predisposing oil palm to toxicity (abiotic effect) and leaving it vulnerable to opportunistic pathogen attacks (biotic effect).

Recent studies using single omics analysis (SOA), such as genomics/meta-genomics, transcriptomics/meta-transcriptomics, proteomics/meta-proteomics, metabolomics, epigenomics, ionomics, and phenomics, have shown that these new techniques can take the etiological studies regarding FY in oil palm to another level [13,14,15]. Costa et al. [13] used a metagenomics approach to rule out the hypothesis that *Phytophthora palmivora* could be the causal agent of FY in Brazil, as has been shown to be the case for Pudricion del Cogollo in Colombia, a disease characterized by the rotting of all the new tissues, preserving the leaves that were formed before infection. This oomycetes is responsible for the first symptoms, and opens doors for several opportunistic pathogens that promote the intensification of the rotting [16,17]. Rodrigues-Neto et al. [14] applied untargeted metabolomics analysis to characterize the leaves of FY asymptomatic and symptomatic oil palm plants, and identified two metabolites (glycerophosphorylcholine and 1,2-dihexanoyl-sn-glycero-3-phosphoethanolamine) with no known direct relation to plant stress, and which are presented as potential biomarkers. Nascimento et al. [15], used a proteomics approach to describe protein alterations associated with FY in oil palm roots, and found enzymes that suggested an anaerobic condition before or during FY. According to [15], their finding suggests that changes in abiotic factors may precede the occurrence of FY, paving the way for opportunistic pathogens.

In the present study, we carried out a comprehensive, large-scale, single- (SOA) and multi-omics integration (MOI) analysis of the metabolome and transcriptome profiles on the leaves of oil palm plants contrasting in terms of FY symptomatology—asymptomatic and symptomatic—and collected in two distinct seasons—dry and rainy. We also performed an analysis of leaf chemical and soil physicochemical composition. The initial goals of such a study were to obtain insights into the possible occurrence and role of oxygen deficiency (hypoxia) in the onset of FY and to search for molecular symptoms (gene- and metabolic pathway-based) that could reveal opportunities for genetic control of this disease of unknown etiology.

## 2. Results

### 2.1. Soil Physicochemical and Leaf Chemical Analysis

In Figure 1, to facilitate the understanding, a design of this study is presented. Figure 2 and Appendix A summarize the results of the ionomics analysis of the soil and oil palm leaves collected in the dry period (DP) and wet period (WP). Asymptomatic and symptomatic plants were compared within each period and between periods.

In the soil, Carbon, Chlorine, Sodium, Phosphorus, and Zinc showed significant differences (*p* ≤ 0.05) between FY asymptomatic and symptomatic plants in the DP, with all showing reduced values in the DP. In soils sampled in the WP, only Calcium and Zinc showed significant differences *p* ≤ 0.05), with increased amounts in the symptomatic plants. Soil organic matter and pH showed significant (*p* ≤ 0.05) lower values in samples from symptomatic plants in the DP but not in the WP. On the other hand, cationic exchange capacity, acidity, and clay showed significant (*p* ≤ 0.05) lower values in soil samples from symptomatic plants. Finally, Ca/CEC (cation-exchange capacity) and K/CEC showed higher values in soil samples from symptomatic plants.

When comparing the soil samples collected in DP and WP conditions, K, Ca, Cu, Fe, Mn, Zn, Na, Al, pH, CEC, OM, silt, and clay showed significantly distinct values (*p* ≤ 0.05). K, Cu, Mn, Zn, Al, OM, and silt showed higher values in WP conditions, while Ca, Fe, Na, pH, CEC, and clay showed lower values in that condition (Figure 2 and Appendix A).

In the case of the leaves from FY asymptomatic and symptomatic oil palm plants in the DP, only Phosphorus and Iron showed significantly distinct values (*p* ≤ 0.05), lower in the latter plants. Meanwhile, Calcium was the only element showing a different (and lower) value in the symptomatic plants in the WP. When comparing the leaf samples collected in DP and WP conditions, N, P, K, Mg, Cl, Cu, and Mn showed significantly distinct values (*p* ≤ 0.05), with N, Mg, and Mn being lower in WP, and P, K, Cl, and Cu higher (Figure 2 and Appendix A).

A principal component analysis (PCA) was performed, revealing the clustering of micro- and macro-nutrients from soil and leaf samples, as well as soil complex and granulometry, accordingly to the collection period (Figure 3). When performing PCA within each period, no clustering of FY asymptomatic and symptomatic plant groups appeared (Appendix A).

### 2.2. Metabolomics Analysis

The Statistical Analysis module of the MetaboAnalyst 5.0 returned 1924, 576, 2469, and 272 peaks for the positive and negative polar and lipidic fractions, respectively, when using the dry period (DP) samples. There were 29 peaks differentially expressed in the positive polar fraction, and 22 in the negative polar fraction. Regarding the lipidic fractions, 29 and 9 were differentially expressed in the positive and negative ones, respectively. Accordingly to the Functional Analysis module of the MetaboAnalyst 5.0, 89 differentially expressed peaks are below the minimal number for functional interpretation using the combined meta-analysis of the mummichog and GSEA pathways. However, when using false discovery rate (FDR) ≤ 0.06, the number of differentially expressed peaks rose to 121, above the minimal number necessary for the functional interpretation analysis. In this case, it was possible to identify 15 differentially expressed metabolites (DEMs) with FDR ≤ 0.05 (Table 1).

In the case of the wet period (WP) samples, the ultra-high performance liquid chromatography and tandem mass spectrometry (UHPLC–MS/MS) statistical analysis returned 1976, 771, 2824, and 461 chromatography peaks for the positive and negative polar and lipidic fractions, respectively. None of them presented differentially expressed peaks using the statistical analysis criteria (FDR ≤ 0.05). When applying the principal component analysis (PCA) to detect any inherent patterns within the data in the DP and WP samples, one could not completely separate the groups between the asymptomatic and the symptomatic plants in all fractions analyzed (Appendix A).

When looking for DEMs between FY asymptomatic oil palm plants from DP and WP, the statistical analysis returned 2267, 836, 2675, and 487 chromatography peaks for the positive and negative polar and lipidic fractions, respectively. Altogether, 2749 differentially expressed chromatography peaks were identified among the asymptomatic plants and subjected to functional interpretation via analysis in the Functional Analysis module of MetaboAnalyst 5.0 (see Section 4), and the combined meta-analysis of the mummichog and GSEA pathways resulted in a list of 303 DEMs (Appendix A). Likewise, for the FY symptomatic plants from DP and WP, the statistical analysis returned 2259, 789, 2549, and 487 peaks for the positive and negative polar and lipidic fractions, respectively. Altogether, 2446 differentially expressed peaks were identified among the symptomatic plants, and subjected to functional interpretation as before, resulting in a list of 259 DEMs (Appendix A). These two groups of DEMs had 179 metabolites in common (Appendix A), while their behavior in the asymptomatic and symptomatic oil palm plants showed a very weak positive correlation (Figure 4).

### 2.3. Transcriptomics Analysis

When comparing the leaf transcriptome of asymptomatic and symptomatic plants collected in the dry period, 274 proteins were differentially expressed (DEPs) when using an FDR ≤ 0.05, and an FC ≠ 1, with 103 upregulated and 171 downregulated. In the wet period, the number of DEPs increased to 1087, with 456 upregulated and 631 downregulated. That amount of differentially expressed proteins—274 and 1087—correspond, respectively, to just 0.63% and 2.50% of all 43,551 proteins present in the reference genome of E. guineensis (Singh et al., 2013). A group of 70 DEPs appeared in both DP and WP, and a correlation analysis was performed to compare their expression profiles under two scenarios, allowing the visualization of 56 proteins with similar expression profiles in the leaves of oil palm plants due to the FY disease, independently of whether it was the dry or wet period (Figure 5).

On the other hand, when comparing the leaf transcriptome of asymptomatic plants in both periods, 6058 proteins were differentially expressed when using an FDR ≤ 0.05, and an FC ≠ 1, with 3071 upregulated and 2987 downregulated. Likewise, when comparing the leaf transcriptome of symptomatic plants in both periods, the number of DEPs was 5426, 2781 upregulated, and 2645 downregulated. A group of 3806 DEPs appeared in both the asymptomatic and symptomatic plants, and a correlation analysis was performed to compare their expression profiles under the two scenarios studied, showing a strong positive correlation (Figure 6). Such transcriptomics results show that the leaf transcriptome of oil palm plants becomes more affected by the change in the environment—from the dry to rainy seasons—than by the presence of the FY disease. Meanwhile, the disease effects were more prevalent in the WP compared to the DP.

After removing the 3806 DEPs that appeared in both the asymptomatic and symptomatic plants from the above-mentioned group of 5426 ones, a set of 1620 that appeared only in the symptomatic plants underwent gene ontology analyses. This set represents those genes/proteins affected by the change from the dry to rainy season, but only in the FY symptomatic plants; there was a direct link between the disease and the environment. In terms of enzyme category, the two most prevalent groups of enzymes were transferases and hydrolases, followed by translocases and oxidoreductases (Figure 7A). In the case of biological process (BP), protein phosphorylation and regulation of transcription, both had approximately 80 positive hits. Protein, ATP, and RNA binding were the most prevalent molecular functions (MF), in that order. Finally, in terms of cellular component (CC), membrane had almost 700 positive hits all together (Figure 7B).

### 2.4. Multi-Omics Integration Analysis

The MOI analysis was employed three times in this study. First, it integrated DEMs and DEPs identified when comparing symptomatic and asymptomatic plants within an specific scenario—dry period or wet period. Here, 15 DEMs and 30 differentially expressed enzymes (out of the 274 DEPs identified when evaluating the differences between symptomatic and asymptomatic plants in the dry period) underwent integration using the Omics Fusion platform. The results revealed that two pathways had five or more of those enzymes differentially expressed; they were Carbon fixation pathways in prokaryotes (p00720) and Methane metabolism (map00680), both with two DEPs and three DEMs. On the other hand, as there were no DEMs identified in the wet period, only the 51 differentially expressed enzymes (out of the 1.087 DEPs identified when evaluating the differences between symptomatic and asymptomatic plants in the wet period) were analyzed using the above-cited platform. The results revealed Glycolysis/Gluconeogenesis (map00010) as the only pathway with five differentially expressed enzymes.

Then, in a second moment, MOI was employed to integrate 96 enzymes—found among the 1620 DEPs that appeared only in the symptomatic plants and underwent gene ontology analyses—and the 80 DEMs present only in symptomatic plants when comparing WP and DP. The results revealed three pathways with ten or more enzymes and metabolites differentially expressed; they were Glycolysis/Gluconeogenesis (map00010), Methane metabolism (map00680), and Cysteine and methionine metabolism (map00270), respectively, with 12, 10, and 10 features (Table 2).

Finally, the group of 5426 DEPs present in the symptomatic plants, and that also including the 3806 DEPs present in the asymptomatic ones, underwent analysis for enzyme selection—there were 320 enzymes—and subsequent integration with the 259 above-mentioned DEMs. In Table 2, a list of 27 metabolic pathways affected only in symptomatic plants, or in both the asymptomatic and symptomatic plants at once, by the change of season—from dry to rainy—is presented. Three pathways had 20 or more enzymes and metabolites differentially expressed; they were Purine metabolism (map00230), Porphyrin and chlorophyll metabolism (map00860), and Phenylpropanoid biosynthesis (map00940), respectively, with 32, 29, and 20 enzymes and metabolites.

## 3. Discussion

Throughout the 50 years since this disease first appeared in Brazil, several initiatives in Brazil and abroad tried to elucidate the etiology of this disease. Not a single one was able to fulfill Koch’s third postulate. With no knowledge about the causal agent(s), nothing was done to develop a diagnostic system or a control measure. For those interested in knowing more about this disease, we encourage you to read [6].

The initial goal of this study was to obtain insights into the possible occurrence and role of oxygen deficiency (hypoxia) in the onset of FY. In that sense, the results showed that the soil underwent much more profound changes in its physicochemical attributes as a function of the season—the change from the dry to rainy season—than as a function of the cultivation of oil palm plants symptomatic or asymptomatic for FY. In the rainy season, the wet soil increased the availability of K, Cu, Mn, Zn, and Al, while decreasing the availability of Ca, Fe, and Na. Concomitantly, it caused a reduction in the pH, clay contents, and cation-exchange capacity, in addition to an increase in the organic matter.

The increase in the availability of cationic micronutrients (Cu, Mn, Zn) and Al in the wet soil, in general, may be related to the decrease in pH. It is known that the lower the soil pH, the greater the availability of cationic micronutrients. Soil pH dropped from a value close to 5.0 (dry period) to 4.1 (wet period). With excess water in the soil, in addition to K, an increase in the availability of other bases, such as Ca and Mg, was expected. However, we saw a decrease in Ca, Fe, and Na, while Mg did not change. The drop in pH may have affected the availability of Ca, Mg, and Na. It is also necessary to consider the changes that occur with the alternation between aerobic and anaerobic conditions, which is reflected in the oxide reduction potential of the soil. For example, organic matter breakdown is slower under anaerobic conditions than under aerobic conditions [18]. And effectively, organic matter increased in the wet period in the oil palm field that supplied the samples for the present study. Additionally, much of the Fe^2+^ formed during reductive dissolution is likely to have been chelated with soil organic matter and, possibly, eluted from the soil; some will have been held by cation exchange in the constituent clay minerals [19].

In summary, the differences in physicochemical attributes of the soil where the sampled oil palm plants grew did not justify the difference in phenotypes—symptomatic and asymptomatic regarding FY—and the same was true for the chemical attributes of the leaves from such plants. So, independently of whether the plants were under hypoxia due to excessive rain and soil waterlogging, no differences regarding leaves and soil attributes justified the distinct FY phenotypes seen among the plants sampled for this study. Would that be different in the leaf metabolome?

Surprisingly, only 15 differentially expressed metabolites (DEM) appeared between symptomatic and asymptomatic plants in the dry period. In the wet period, it was even worst, with no DEM found when using FDR ≤ 0.05 as the statistical analysis criteria. In synthesis, the metabolomes from the leaves of oil palm plants did not present pathways highly affected by the disease. The same was not true for those cases where the metabolomes were from plants with the same phenotype but at different seasons. Here, it was possible to select 80 metabolites that only differentially expressed in the symptomatic plants, not in the asymptomatic; and those metabolites allowed the identification of four pathways affected in sick plants by the rainy season; they were Steroid biosynthesis (map00100), Carbon fixation pathways in prokaryotes (p00720), Carotenoid biosynthesis (map00906), and Purine metabolism (map00230). Such pathways represent a direct link between the disease and the environment and might help to understand, at the molecular level, the intensification of the FY symptoms seen in the leaves in the rainy season. It is common sense that in the rainy season, the visual FY symptoms intensify (Denpasa’s staff—Dendê do Pará S/A company—www.denpasa.com.br (accessed on 30 June 2023), personal communication).

As mentioned in Bittencourt et al. [20], untargeted metabolomics allows the search for novel metabolic perturbations in various biological systems. However, as seen in the present study, when using the profile of hundreds or thousands of peaks with varying chemical properties, just a few dozen metabolites are identified. The reasons behind this are the still limited capacity to identify novel compounds of interest and the need for advanced and more robust databases [21]. Would that be different in the leaf transcriptome?

RNA-Seq uses deep-sequencing technologies to characterize the transcriptome profiling of a cell, a tissue, an organ, or even the entire organism, and provides a far more precise measurement of levels of transcripts and their isoforms than any other methods [22]. Accordingly to Wang et al. [22], characterizing the transcriptome allows us to catalog all types of transcripts present in that cell/tissue/organ/organism at a specific moment, and to quantify the changes in expression levels of each transcript under distinct scenarios; moreover, it allows us to determine the transcriptional structure of genes, in terms of their start sites, 5′ and 3′ ends, splicing patterns and other post-transcriptional modifications.

RNA-seq became a powerful tool to study host–pathogen interactions, enlarging the horizon of opportunities for the development of early diagnosis tools, as well as for the identification of candidate genes to be employed in the development of improved genotypes resistant to a specific disease [23,24,25,26]. In the present study, RNA-Seq allowed not only the identification of genes/proteins differentially expressed in the leaves of FY symptomatic oil palm plants in comparison to the asymptomatic ones—either in the dry or rainy seasons— but also the identification of those differentially expressed in plants with similar FY-based phenotypes between seasons.

A set of 56 genes/proteins differentially expressed in the leaves of oil palm plants symptomatic for FY compared to asymptomatic ones, either in the dry or rainy seasons, was selected after transcriptomics analysis. They are molecular symptoms in the plant directly linked to the disease, which are positively (33 proteins) or negatively (23 proteins) expressed in the symptomatic plants—in comparison with the asymptomatic ones—independently of the season. Here, three of them underwent discussion, and the remaining 53 genes/proteins will undergo further analysis, and the results will be reported in the future. The protein most negatively regulated among those 56 selected (Figure 7) codes for a ribosomal protein large (RPL) subunit. RPLs are the components of the ribosome machinery and, to a certain extent, are required for protein synthesis. The ribosomal proteins names follow the subunit of the ribosome to which they belong—the small (S1 to S31) and the large (L1 to L44) [27].

This oil palm RPL belonged to the Ribosomal protein L19 protein family (IPR001857) and the Ribosomal protein L19 homologous superfamily type (IPR038657). In symptomatic plants, the expression level of that gene was reduced to 1.7% and 0.27% of the initial level seen in the asymptomatic plants in the dry and rainy seasons, respectively. Nagaraj et al. [28] showed that when NbRPL19 was silenced in *Nicotiana benthamiana*, the non-host resistance became compromised, and the same was true in *Arabidopsis* mutants for AtRPL19. More recently, Ramu et al. [29] reported that RPL10-silenced *N. benthamiana* plants showed compromised disease resistance against the non-host pathogen *Pseudomonas syringae* pv. *tomato* T1.

Non-host resistance (NHR) is a safety barrier that protects plants from a large and diverse array of potential phytopathogens. Non-host species present an innate immunity that cannot be overcome by potential pathogenic microbes, resulting from a series of physical, chemical, and inducible defenses. NHR is a very durable type of resistance, which has raised great interest everywhere regarding its genetic basis and functionally transferring it to plant species of commercial interest [30,31,32].

Besides being linked to resistance against non-host pathogens, RPL19 also showed high RNA-chaperone activity in a trans-splicing assay where the pre-mRNA of the thymidylate synthase (td) gene containing a group I intron was spliced into two halves [33]. Kovacs et al. [34] showed that RPL19 from *E. coli* is partially unstructured and/or has molten globule-like characteristics once without the support of rRNA, and exhibited potent chaperone activity with the substrates alcohol dehydrogenase (ADH) and lysozyme in three different chaperone assays. Finally, Gorelova et al. [35] demonstrated that one of the bifunctional dihydrofolate reductase/thymidylate synthase (DHFR-TS) isoforms of *A. thaliana* (*At2g21550*) operates as an inhibitor of its homologs, regulating DHFR and TS activities. Such regulation affects folate abundance. Gorelova and colleagues also proposed a novel function of folate metabolism in plants, which is the maintenance of the redox balance by contributing to NADPH production through the reaction catalyzed by methylenetetrahydrofolate dehydrogenase, thus allowing plants to cope with oxidative stress [35].

*At2g21550* codes for NP_001324557.1, and using that protein sequence once to Blastp against the reference genome of oil palm [36], six positive hits appeared. Five were from a gene (LOC105048636) in chromosome seven. The former did not differentially express in the transcriptome of oil palm plants, but the latter did. There, the latter showed reduced expression in the rainy season, compared with the dry season, in both the asymptomatic and symptomatic plants, but was not differentially expressed in symptomatic plants compared with asymptomatic plants in the dry or rainy seasons. According to InterPro [37], XP_029121477.1 (coded by LOC105048636) is a representative of the bifunctional dihydrofolate reductase/thymidylate synthase (IPR012262) family, is positive for the dTMP biosynthetic process (GO:0006231), the one-carbon metabolic process (GO:0006730), the tetrahydrofolate biosynthetic process (GO:0046654), the biological process, and for thymidylate synthase activity (GO:0004799), dihydrofolate reductase activity (GO:0004146), and transferase activity, for transferring one-carbon groups (GO:0016741), and for molecular function. Altogether, the results in the present study show that the negative regulation of XP_029121477.1 is due to the changes in the environment—from dry to rainy season— and is not linked to the change in phenotype from asymptomatic to symptomatic.

This specific oil palm RPL is under the regulation of twelve genes, according to a gene regulatory network available in our lab [38] featuring epigenetic regulators and transcription factors from the oil palm genome and built based on the strategy reported by McCoy et al. [39]. Such analysis was performed by applying GENIE3 to mine 306 public oil palm transcriptome datasets. Such a study used 1333 unique regulators and 27,642 target genes from the oil palm reference genome [36]. The expression profiles of the twelve genes showed seven of them not present or not differentially expressed in the four scenarios evaluated in the present study. The scenarios are: (a) symptomatic vs. asymptomatic in the dry period; (b) symptomatic vs. asymptomatic in the wet period; (c) wet vs. dry period in symptomatic plants; and (d) wet vs. dry in asymptomatic ones. Among the remaining five, three were not differentially expressed in the two first scenarios but were positively regulated in the two last. Finally, two were only differentially expressed in the third scenario; one positive and one negative. In summary, such results do not allow us to point out any direct effect of any of those 12 regulatory genes in the changes of expression observed in *EgRPL19-2* in the two first scenarios.

The whole genome sequence of the American oil palm (*E. oleifera*) [40] revealed four positive hits for ribosomal protein L19-2. A Blastp analysis of those four positive hits against the oil palm reference genome [36] allowed the identification of two proteins in the African oil palm with expression profiles highly different from each other in the four above-mentioned scenarios. The *E. guineensis* RPL19-2 protein, whose expression level was highly reduced in FY-symptomatic plants in the dry and rainy seasons, compared to the asymptomatic ones, has very high identification with three of the above-mentioned positive hits for ribosomal protein L19-2 protein in *E. oleifera*, a species known to be resistant to FY [6].

Among those 33 proteins down-regulated there were two WAK-like proteins (WAKLs). Wall-associated kinases (WAKs) and WAKs-like proteins (WAKLs) belong to a plant-specific subfamily of the receptor-like kinase family (IPR045274), and some of them have been implicated in resistance to bacterial and fungal diseases [41,42]. When investigating the defense role of a pathogen-induced WAK gene from wheat chromosome 7D, designed as *TaWAK7D*, Qi et al. [41] suggested that such a gene positively participates in the defense against infection by the soilborne and necrotrophic fungus *Rhizoctonia cerealis*, through activating the expression of several pathogenesis-related (PR) genes, including *Chitinase3*, *Chitinase4*, *PR1*, *PR17* and *β-1,3-Glucanase*.

Plants have either plasma membrane-localized receptor kinases (RKs) or receptor-like proteins that perceive pathogen- or microbe-associated molecular patterns (PAMPs/MAMPs), as well as damage-associated molecular patterns (DAMPs). Recognizing such patterns triggers immunity, which contributes to basal immunity to adapted pathogens and NHR to non-adapted pathogens by the induction of both local and systemic immune responses [43].

The expression levels of the two WAKLs proteins—differentially expressed in the leaves of FY-symptomatic oil palm plants in the present study—were reduced to 25–45% of the initial level seen in the asymptomatic plants. The genes that code XP_010934766.2 (LOC105054847), a wall-associated receptor kinase 2-like protein, and XP_010934767.1 (LOC105054848), a putative wall-associated receptor kinase-like 16, are located side by side in chromosome 12 of the oil palm reference genome [36]. According to InterPro [37], they are representative of the Receptor-like kinase WAK-like (IPR045274) family, positive for protein phosphorylation (GO:0006468) and for cell surface receptor signaling (GO:0007166), for biological process, and ATP binding (GO:0005524), protein kinase activity (GO:0004672), calcium ion binding (GO:0005509), polysaccharide binding (GO:0030247), and molecular function. Moreover, they showed 85% of identity measured across approximately their entire amino acid sequence.

Again, the gene regulatory network in our lab [38] showed that no unique regulator regulates the LOC105054848 (XP_010934767.1) gene/protein, and the LOC105054847 (XP_010934766.2) might be under the regulation of 16 genes. The transcriptome generated in the present study revealed that 13 genes out of the 16 were not present in the transcriptome or did not differentially express in any of the four scenarios evaluated. Among the remaining three, two were negatively and one positively regulated in the last scenario. Finally, two were differentially expressed only in the third scenario; one positively and one negatively. In summary, such results also do not allow to us to point out any direct effect of any of those 16 regulatory genes in the changes of expression observed in LOC105054847 in the first two scenarios.

The pathway-based MOI analysis performed in this study using the Omics Fusion platform brought together enzymes (from transcriptomics studies) and metabolites that expressed differentially in the leaves of oil palm plants under distinct conditions (dry and wet periods or dry and rainy seasons). First, the analysis integrated differentially expressed metabolites and enzymes found only in symptomatic plants and then those found in both asymptomatic and symptomatic plants. By doing that, our results revealed those pathways affected by the environment—independently of the FY phenotype—but also allowed us to map those enzymes and metabolites that play a role only in the symptomatic plants. For instance, in the case of Purine metabolism (map00230), five metabolites (out of seventeen) appeared only in the symptomatic plant, and the same was true for four enzymes (out of fifteen). Moreover, two of the remaining twelve metabolites had distinct qualitative expression profiles, while all remaining proteins had similar qualitative expression profiles in asymptomatic and symptomatic plants. At the same time, this allowed the identification of metabolites and enzymes with similar qualitative expression profiles but with differences in the quantitative one. For instance, Inosine monophosphate (IMP—C00130) was positively regulated 113 times in the symptomatic plants and only 3 times in the asymptomatic ones; meanwhile, Inosine diphosphate (IDP—C00104) was positively regulated 88 times in the symptomatic plants, and negatively in only 7% of the asymptomatic ones (Table 2, Table 3 and Table 4).

The groundwater at the oil palm field where the plants sampled in the present study were growing was almost on the soil surface during the rainy season (Figure 8E), indicating that those plants were growing in waterlogged soil and likely experiencing oxygen deficiency (hypoxia), which was possibly affecting their root system. The soil physical–chemical and leaf chemical analyses pointed out differences due to changes in the environment but not due to the FY phenotype. It is common sense that in the rainy season, the visual symptoms of this disease intensify, and the results of the present study show that the number of differentially expressed genes/proteins and metabolites is much higher when one compares plants with the same phenotype in different seasons than between symptomatic and asymptomatic plants in a specific season.

Oxygen deficiency in plants affects several metabolic pathways, and under such conditions substantial changes in the expression levels of transcripts, proteins, and metabolites have been observed [44,45]. However, the initial cellular response to a decrease in O_2_ availability, regardless of whether the species is tolerant or not, is the promotion of the anaerobic metabolism of pyruvate, which is highly conserved in plants and animals [46,47]. Perhaps for this reason, glycolysis and the Krebs cycle are the most-studied metabolic pathways under conditions of hypoxia/anoxia, as pyruvate is the end product of the first and the initial substrate of the second.

The MOI results from this present study showed that all three metabolites identified in the Glycolysis/Gluconeogenesis (map00010) pathway were differentially expressed only in the symptomatic plants, and all were positively regulated in the rainy season. Beta-D-Fructose 6-phosphate (C05345) was the metabolite with the top increase in expression level, 12×, followed by beta-D-Fructose 1,6-bisphosphate (C05378), with 5×, and Acetaldehyde (C00084), with 3×. In the case of the TCA Cycle (map00020) pathway, just one (S)-Malate (C00149) was differentially expressed only in symptomatic plants, with a 6× increase in expression level. Isocitrate (C00311), cis-Aconitate (C00417), and 2-Oxoglutarate (C00026) are negatively regulated in asymptomatic plants and positively regulated in symptomatic. In terms of proteins, Malate dehydrogenase and Succinate--CoA ligase [ADP-forming] subunit beta (mitochondrial) experienced a 50% increase in expression, and phosphoenolpyruvate carboxykinase (ATP) expression level was reduced to 25% of the level in symptomatic plants during the rainy season, without any change in expression level in the asymptomatic plants.

## 4. Materials and Methods

### 4.1. Soil and Leaf Samples—Collection and Chemical and Physicochemical Analysis

The soil and plant material used in this study came from a commercial oil palm plantation belonging to Denpasa—Dendê do Pará S/A company (www.denpasa.com.br, accessed on 30 June 2023) located in Santa Bárbara do Pará, state of Pará, northern Brazil (1°13′25″ S and 48°17′40″ W, 21 m above sea level). This oil palm field started in 2011 and has shown a high incidence of FY, with about 19% of plants affected in 2021, according to Denpasa’s staff (personal communication) (Figure 8).

Soil and leaf samples were collected from asymptomatic and symptomatic plants in the intermediate stages of the disease [48] in two distinct periods: in October 2021—the dry period (DP)—and in June 2022—the wet period (WP). The selected asymptomatic individuals had never shown symptoms of AF, according to Denpasa’s staff (personal communication). The same individuals were sampled in the DP and WP, with only one symptomatic plant replaced (Appendix A).

Soil samples collected from three equidistant points around the plant stem—one meter from it and at 10 cm in diameter and 30 cm deep holes—were homogenized and stored in a plastic bag. Six asymptomatic and six symptomatic plants were sampled in DP, totaling twelve samples, and eight in WP, per treatment, totaling sixteen. Before being sent for analysis, all soil samples were dried at room temperature. Leaves from six asymptomatic and six symptomatic plants were sampled in DP and WP, totaling 24 samples. Leaf samples were dried in an oven at 65 °C, ground using a Wiley mill (Model TE 680, Tecnal, Piracicaba, SP, Brazil), and passed through a 1 mm (20-mesh) sieve. Soil and leaf samples underwent analysis at Soloquímica (www.soloquimica.com.br)—DP samples—and at Terra Análises para Agropecuária (www.laboratorioterra.com.br)—WP samples.

### 4.2. Experimental Design and Statistical Analysis

A completely randomized design was adopted to investigate the influence of two factors and their interaction on the selected soil and leaf variables. The four groups (‘treatments’) were then constituted of combinations of the state of the soil of the experimental area (DP and WP) and the status of plants regarding the FY disorder (symptomatic or asymptomatic). Analyses considered two methods: (a) a separate analysis for each period state, and (b) a conjoint analysis using data from both periods.

We investigated the influence of plant state (symptomatic or asymptomatic) on each of the response variables classified into four groups: soil/leaf macro and micronutrient, soil sorption complex, and soil granulometry. The one-way analysis of variance (ANOVA) was used in this study. Because the factor plant status has only two levels, the ANOVA F-test was applied to compare the factor means. Then we also performed a principal component analysis (PCA) for each of the four groups of variables, as a graphical complementary way of investigating whether plant status was a factor for separating plant groups.

The two-way repeated measures ANOVA quantified the influence of period status and plant status, and their interaction on each response variable. The means corresponding to each period status within plant status were compared via the F-test for contrasts and the means for each plant status were determined by ANOVA F-tests. The PCA was also performed for each of the four groups of variables, but using measurements made during the wet and dry periods and including all samples. We used those measurements that were present in both periods.

In both situations, the statistical software SAS/STAT^®^ was employed; PCA was carried out using the PRINCOMP Procedure for PCA analyses, and the GLM (General Linear Model) Procedure for the ANOVAs (SAS Institute Inc., 2020, Tokyo, Japan). Data were standardized before running PCAs to avoid conflicts due to the different magnitudes of the response variables within each group. The significance level adopted for ANOVAs was 0.05.

### 4.3. Transcriptomics Analysis

The second apical leaf, counted after the arrow leaf, was collected for transcriptomic analysis, and a total of six leaflets from each side of the intermediate portion of the leaf were harvested and sectioned into 10 cm portions from their base. The leaflet sections were stored in RNA later™ solution (Invitrogen, Waltham, MA, USA) on ice, transported to the laboratory, removed from the RNA later™ solution, and kept at −80 °C until extraction. Six biological replicates were collected from symptomatic and six from asymptomatic plants in DP and WP, totaling 24 samples.

Total RNA was isolated from oil palm leaves using the Qiagen RNeasy^®^ Plant Mini kit (QIAGEN, Redwood City, CA, USA), following the manufacturer’s protocol. The quantity and quality of RNA were measured using a Nanodrop Qubit 2.0 fluorometer (Life Technologies, Carlsbad, CA, USA). Library preparation and RNA-Seq were performed by the GenOne Company (Rio de Janeiro, RJ, Brazil) using an Illumina platform and the paired-end strategy.

All RNA-Seq analyses were performed using the OmicsBox platform, version 2.2.4 [49]. We used FastQC [50] and Trimmomatic [51] for quality control, read filtering, and removal of low-quality bases. The oil palm reference genome [36]—files downloaded from NCBI (BioProject PRJNA192219; BioSample SAMN02981535) in October 2020—was used to align the RNA-Seq data using standard OmicsBox version 2.2.4 parameters, through the STAR software [52].

HTSeq version 0.9.0 was used to quantify gene or transcript expression [53], applying the standard parameters of OmicsBox version 2.2.4. Paired differential expression analysis between experimental conditions (symptomatic vs. asymptomatic) was performed using edgeR version 3.28.0 [54], applying a simple design and exact statistical test without filtering for low-count genes.

### 4.4. Metabolomics Analysis

The leaf samples for metabolomics analysis were collected simultaneously and using the same criteria as for the transcriptomics samples, following the “split-sample data” strategy. Six biological replicates were collected for both symptomatic and asymptomatic individuals in DP and WP, resulting in a total of 24 samples.

Before solvent extraction, all samples underwent grounding in liquid nitrogen. We employed a well-established protocol [55,56] to extract the metabolites in three phases (polar, non-polar, and protein pellet) from aliquots of 50 mg of ground tissue. The solvents used were methanol grade UHPLC, acetonitrile grade LC-MS, formic acid grade LC-MS, and sodium hydroxide ACS grade LC-MS, all from Sigma-Aldrich, with water treated in a Milli-Q system from Millipore.

The analytical method ultra-high performance liquid chromatography and tandem mass spectrometry (UHPLC–MS/MS) was used in this study, with the UHPLC system (Nexera X2, Shimadzu Corporation, Kyoto City, Japan) equipped with a C8 reverse-phase column from Waters Technologies (Acquity UPLC HSS T3, 1.8 μm, 2.1 by 150 mm at 35 °C). Solvent A was 0.1% (*v*/*v*) formic acid in water and solvent B was 0.1% (*v*/*v*) formic acid in acetonitrile/methanol (70:30, *v*/*v*). The gradient elution used, with a flow rate of 0.4 mL min^−1^, was as follows: 0–1 min isocratic, 0% B; 1–3 min, 5% B; 3–10 min, 50% B; 10–13 min, 100% B; 13–15 min isocratic, 100% B; then five minutes re-balancing to the initial conditions. The column temperature was set at 40 °C.

High-resolution mass spectrometry was used for detection (MaXis 4G Q-TOF MS, Bruker Daltonics), equipped with an electrospray source in positive (ESI-(+)-MS) and negative (ESI-(−)-MS) modes. The settings of the mass spectrometer were as follows: capillary voltage, 3800 V; dry gas flow, 9 L min^−1^; dry temperature, 200 °C; nebulizer pressure, 4 bar; and final plate offset, 500 V. The rate of acquisition spectra was 3.00 Hz, mass range *m*/*z* 70–1200 for the polar fraction analysis and *m*/*z* 300–1600 for the lipidic fraction. For external calibration of the equipment, we used a sodium formate solution (10 mM HCOONa solution in 50:50 *v*/*v* isopropanol and water containing 0.2% formic acid), injected through a six-way valve at the beginning of each chromatographic run. Ampicillin ([M + H] + *m*/*z* 350.1186729 and [M − H]^−^ *m*/*z* 348.1028826) was the internal standard for later peak normalization of data analysis.

The DataAnalysis 4.2 software (Bruker Daltonics, Bremen, Germany) was the first used to analyze the raw data from UHPLC-MS, as mzMXL files. Pre-processing of data was performed using XCMS Online [57,58], including peak detection, retention time correction, and alignment of the metabolites. CentWave was used for peak detection (maximum peak width, 20 s; Δ*m*/*z* = 10 ppm; minimum peak width, 5 s). For the alignment of retention times, the parameters were mzwid = 0.015, minfrac = 0.5 and bw = 5. The unpaired parametric *t*-test (Welch *t*-test) was used for the statistical analysis at the pre-processing stage.

Initially, the pre-processed data (csv file) underwent analysis in the Statistical Analysis module of the MetaboAnalyst 5.0 [59], using the Pareto method as scaling [60]. Then, the differentially expressed peaks (DEPs)—those passing the criteria of false rate discovery (FDR) ≤ 0.05 and Log_2_(fold change [FC]) ≠ 1—were selected and submitted to the Functional Analysis module, applying the following parameters: molecular weight tolerance of 5 ppm; mixed ion mode; joint analysis using both the mummichog [61] and Gene Set Enrichment Analysis (GSEA) [62] algorithms, and the latest KEGG version of the *Oryza sativa* pathway library. The *p*-value cutoff from the mummichog algorithm was at 1.0 × 10^−5^.

DEPs with two or more matched forms were observed. In those cases, the mass error was the criteria for the feature selection, keeping the smallest. Then, KEGG IDs with two or more features (m.z) were also observed; and, again, the smallest mass error was the criteria for the feature selection. Finally, the KEGG IDs of the matched compounds—one KEGG ID per m.z—were submitted to the pathway analysis module for visualization through integrating enrichment and pathway topology analysis [63]. The parameter sets were the hypergeometric test and the latest KEGG version of the *O. sativa* pathway library.

### 4.5. Correlation and Integratomics Analysis

DEPs and DEMs underwent correlation analysis under two distinct scenarios, symptomatic × asymptomatic plants and dry × wet periods. First, to check for the data distribution, the Data Overview module of Omics Fusion [64], the web platform for integrative analysis of omics data, was used, and then the Scatter Plot one for the correlation analysis between the sets of data—a pairwise combination of the different scenarios evaluated. The input data was the Log_2_(FC) from the DE molecules obtained from the single-omics analysis.

The DEPs and DEMs identified underwent a pathway-mapping approach of integration using the Omics Fusion platform [64]. Before the integration, the NCBI accession of enzymes was converted to UniProt ID. Thus, the input data used were the UniProt accession ids for transcriptomics and KEGG ids for metabolomics. The data underwent enrichment through several databases (EMBL—www.embl.org (accessed on 30 June 2023), KEGG—www.genome.jp/kegg (accessed on 30 June 2023), NCBI—www.ncbi.nlm.nih.gov (accessed on 30 June 2023), and UniProt—www.uniprot.org (accessed on 30 June 2023), and then the module “KEGG feature distribution” was used to map these omics data in known pathways—www.genome.jp/kegg/annotation (accessed on 30 June 2023)).

## 5. Conclusions

This study aimed to obtain insights into the possible occurrence and role of oxygen deficiency (hypoxia) in the onset of Fatal Yellowing (FY), a disease of unknown etiology that limits the oil palm industry in Brazil. Soil and leaf samples from asymptomatic and symptomatic plants in the intermediate stages of the disease were collected in two distinct periods: in October 2021—the dry season—and in June 2022—the rainy season. The changes observed in the physicochemical attributes did not allow for the discrimination of plants asymptomatic or symptomatic for this disease, not even in the rainy season, when the soil became waterlogged. The same was true for the chemical attributes and the metabolome profiles of the leaves. Only transcriptome profiles of the leaves allowed the identification of molecular symptoms able to distinguish symptomatic from asymptomatic plants, independently of the season—dry or rainy. A set of 56 proteins/genes, negatively or positively regulated in symptomatic plants compared to the asymptomatic ones, resulting from this study, is undergoing additional analysis, aiming at a broad in silico functional annotation and the validation of the RNA-Seq expression profile employing qPCR analysis.

Altogether, the single-omics analysis (SOA) performed in the present study allowed the identification of 320 enzymes (from the transcriptome analysis) and 254 metabolites on the leaves of oil palm plants that underwent multi-omics integration (MOI) analysis. Such a set was composed of enzymes and metabolites differentially expressed in asymptomatic and symptomatic plants in the rainy season—waterlogged soil—compared to the dry season, plus those differentially expressed only in the symptomatic ones. Such an MOI analysis produced a list of 27 metabolic pathways affected by the change from dry to rainy season, with at least ten enzymes and metabolites differentially expressed. Starting from the premise that the visual FY symptoms intensify in the rainy season, we postulate that a closer look at such pathways might reveal insights into the role of hypoxia in the symptom intensification of FY.

Finally, the closer analysis of three out of the fifty-six proteins/genes selected employing transcriptomics analysis under four distinct scenarios strongly points to the following postulate: the oxygen deficiency (hypoxia) experienced by the oil palm plants for long periods of the year promotes stress in the roots of those plants and triggers, directly or indirectly, a cascade of events that breaks some of the safety barriers that protect plants from a large and diverse array of potential phytopathogens. Breaks in the non-host resistance to non-adapted pathogens—as suggested by the strong negative regulation of EgRPL19-2 in both seasons—as well as in the basal immunity to adapted pathogens, as pointed out by the negative regulation of this gene and of two WAKs-like proteins belonging to a plant-specific subfamily of the receptor-like kinase family (IPR045274), are the initial basis for such a postulate. By doing this, it creates the possibility for opportunist microorganisms in the soil to infect the plant and promote this bud-rot type of disease. Whether a specific opportunistic pathogen is prevalent or not still needs further evaluation, although we also postulate that this might not be the case.

Why do we find asymptomatic plants surrounded by symptomatic ones after spending a decade in the same conditions? The variability in the expression profiles of those three genes—and several others among the 53 remaining for further characterization—within this plant species, but also in the American oil palm (*E. oleifera*) population, and inside the populations of inter-specific hybrids between these two species, can pave the way to answering this question and identify bio-markers for the selection of oil palm plants resistant to the Fatal Yellowing disease. Mapping the differences in the promoter sequence of such genes, as well as those between them and their orthologs in the American oil palm, might help developing gene editing strategies able to protect such genes from the cascade of events triggered by the abiotic stress, and maintain the safety barriers raised and strong.

## Figures and Tables

**Figure 1 ijms-24-12918-f001:**
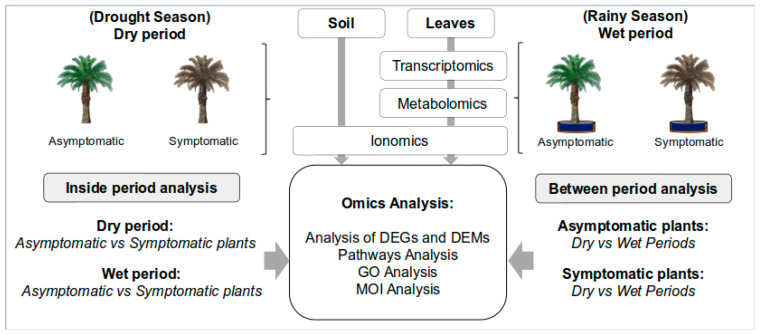
Experimental design for sample collection, and the general work-flow of the strategy of the analysis carried out in four scenarios—symptomatic vs. asymptomatic in the dry period; symptomatic vs. asymptomatic in the wet period; wet vs. dry period in symptomatic plants; and wet vs. dry in asymptomatic ones.

**Figure 2 ijms-24-12918-f002:**
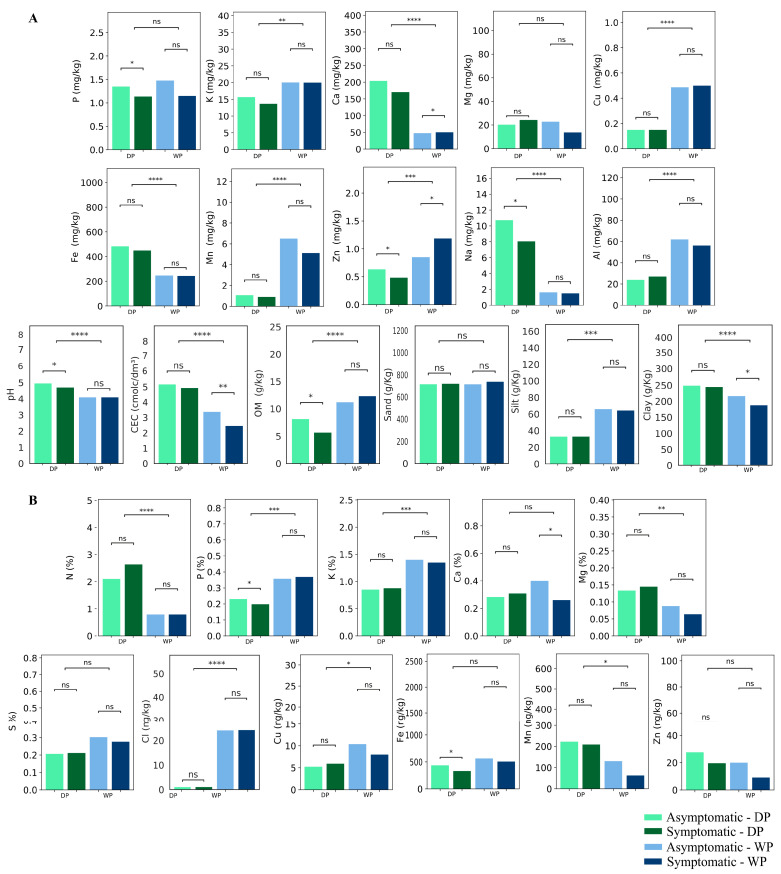
Physical–chemical characteristics of the soil (**A**) and chemical from the leaves (**B**) of FY asymptomatic and symptomatic oil palm plants sampled in two time-points: Dry period—DP and Wet period—WP. The ns means non-significant. The asterisks indicate a significant difference between the two groups (*t*-test). * *p* ≤ 0.05; ** *p* ≤ 0.01; *** *p* ≤ 0.001; **** *p* ≤ 0.0001.

**Figure 3 ijms-24-12918-f003:**
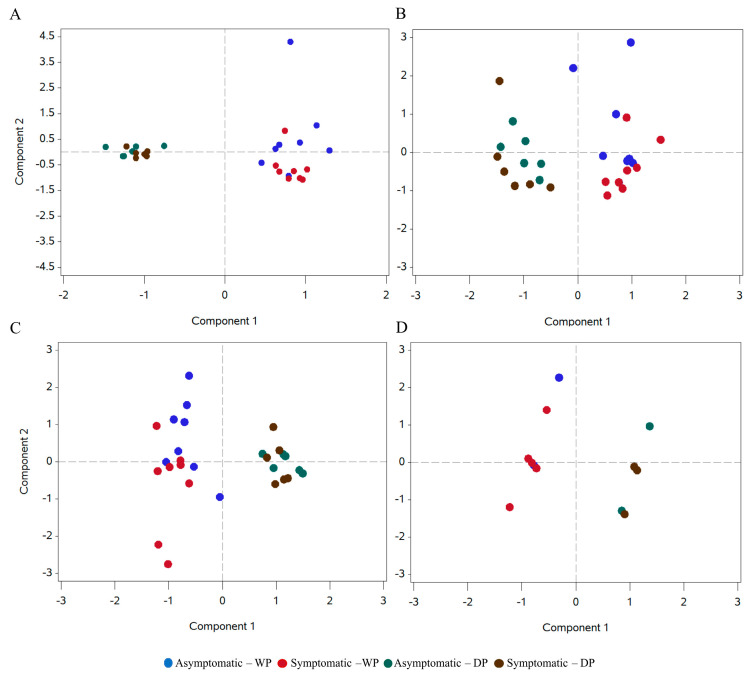
Principal component analysis (PCA) of the micro− and macro−nutrients from soil (**A**) and leaves (**B**), assorted complex (**C**) and granulometry (**D**) including all samples of FY asymptomatic and symptomatic oil palm plants sampled in Dry and Wet period.

**Figure 4 ijms-24-12918-f004:**
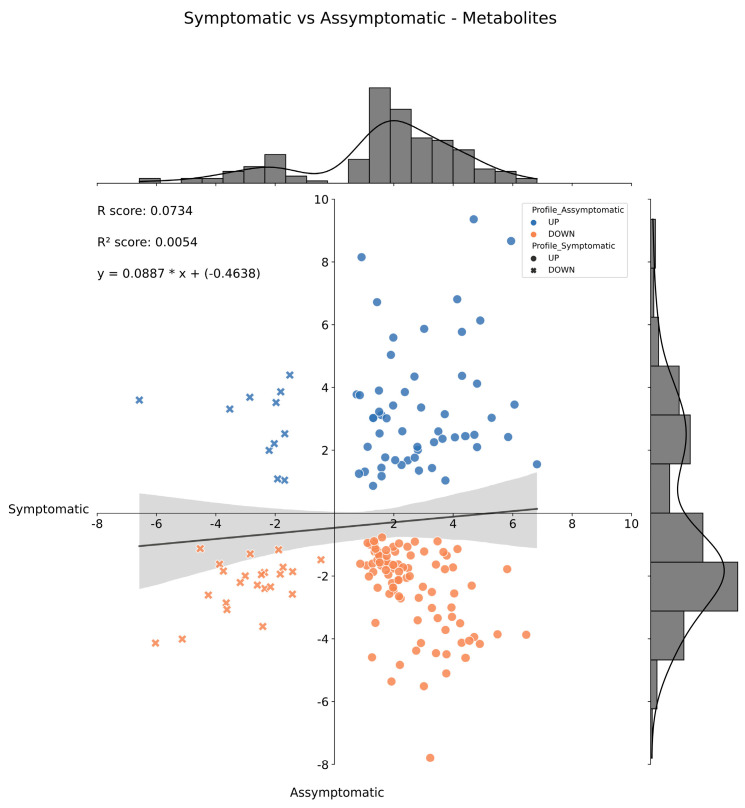
Histogram and correlation analysis of the Log_2_(FC) of common differentially expressed metabolites from Dry vs. Wet periods of FY asymptomatic and symptomatic plants by pairwise comparison. Dots represent metabolites positively regulated in FY symptomatic plants; x’s represent metabolites negatively regulated in FY asymptomatic plants. Blue dots and x’s represent metabolites positively regulated in FY symptomatic plants, and orange dots and x’s represent metabolites negatively regulated under FY asymptomatic plants. FC = Fold Change.

**Figure 5 ijms-24-12918-f005:**
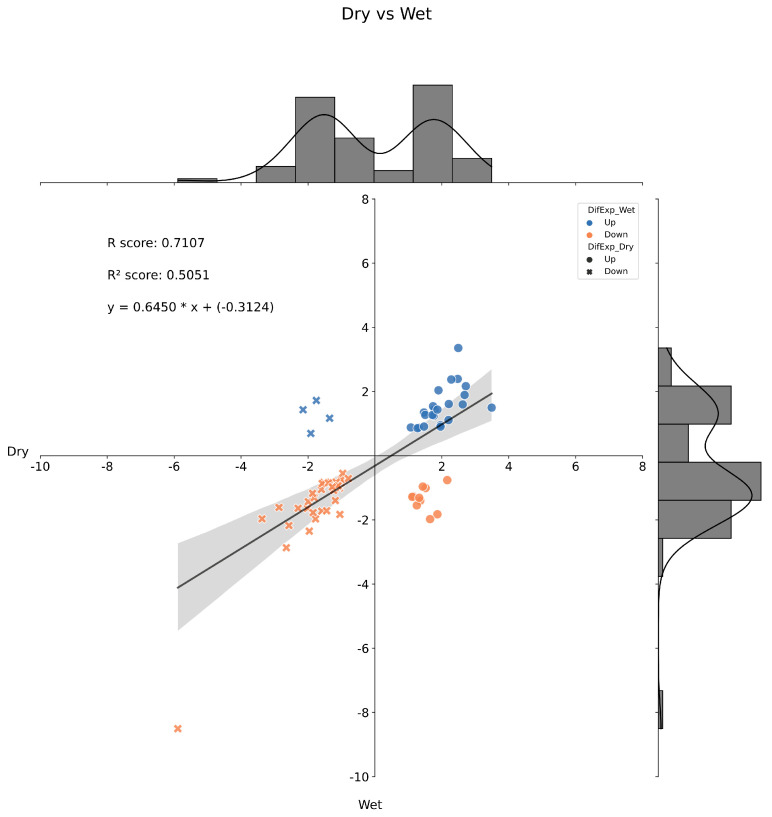
Histogram and correlation analysis of the Log_2_(FC) of common differentially expressed metabolites by pairwise comparison from Dry and Wet periods of FY asymptomatic vs. symptomatic analysis. Dots represent metabolites positively regulated in the Dry period; x’s represent metabolites negatively regulated in the Wet period. Blue dots and x’s represent metabolites positively regulated under the Dry period, and orange dots and x’s represent metabolites negatively regulated in the Wet period. FC = Fold Change.

**Figure 6 ijms-24-12918-f006:**
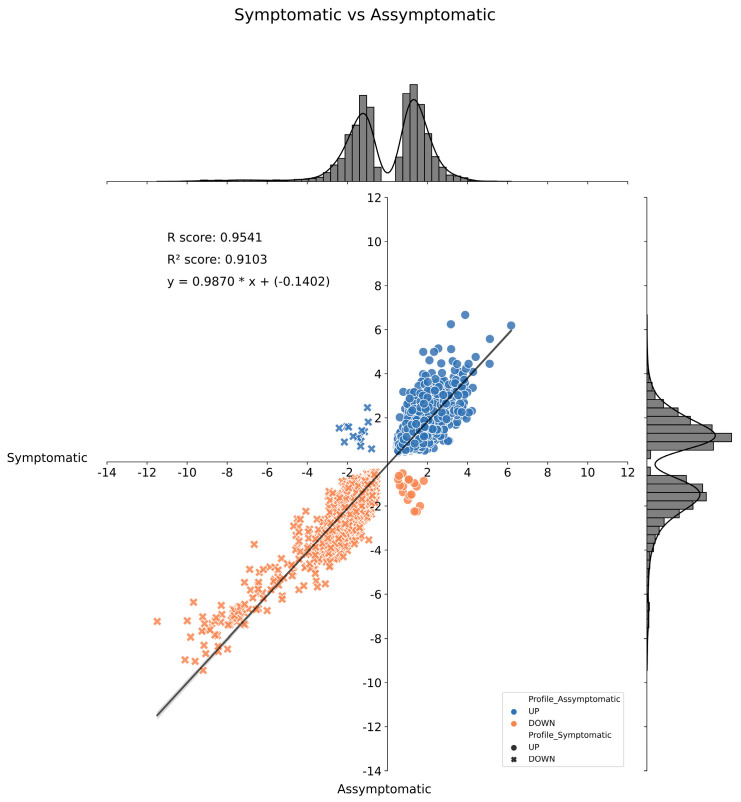
Histogram and correlation analysis of the Log_2_(FC) of common differentially expressed genes by pairwise comparison from Dry vs. Wet periods of FY asymptomatic and symptomatic analysis. Dots represent genes positively regulated in FY symptomatic plants; x’s represent genes negatively regulated in FY asymptomatic plants. Blue dots and x’s represent metabolites positively regulated in FY symptomatic plants, and orange dots and x’s represent metabolites negatively regulated in FY asymptomatic plants. FC = Fold Change.

**Figure 7 ijms-24-12918-f007:**
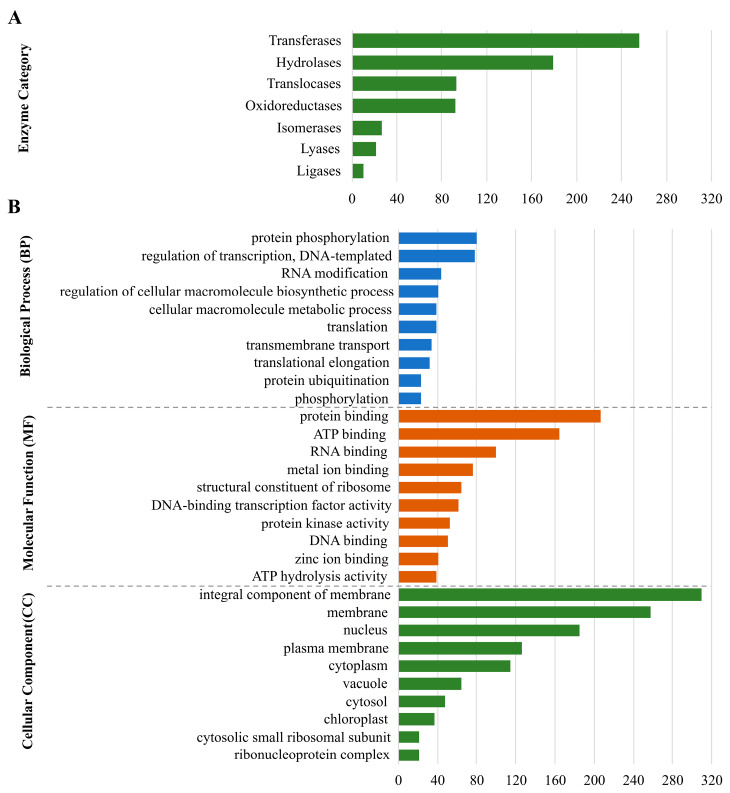
Amount of full-length transcripts present only in FY-affected oil palm based on Gene Ontology (GO) annotation from 1602 differential expresses genes/proteins that appear only in FY symptomatic oil palm plants; enzyme classification (**A**) and biological process, molecular function, and cellular component (**B**). Only the ten most populated groups per GO term are shown.

**Figure 8 ijms-24-12918-f008:**
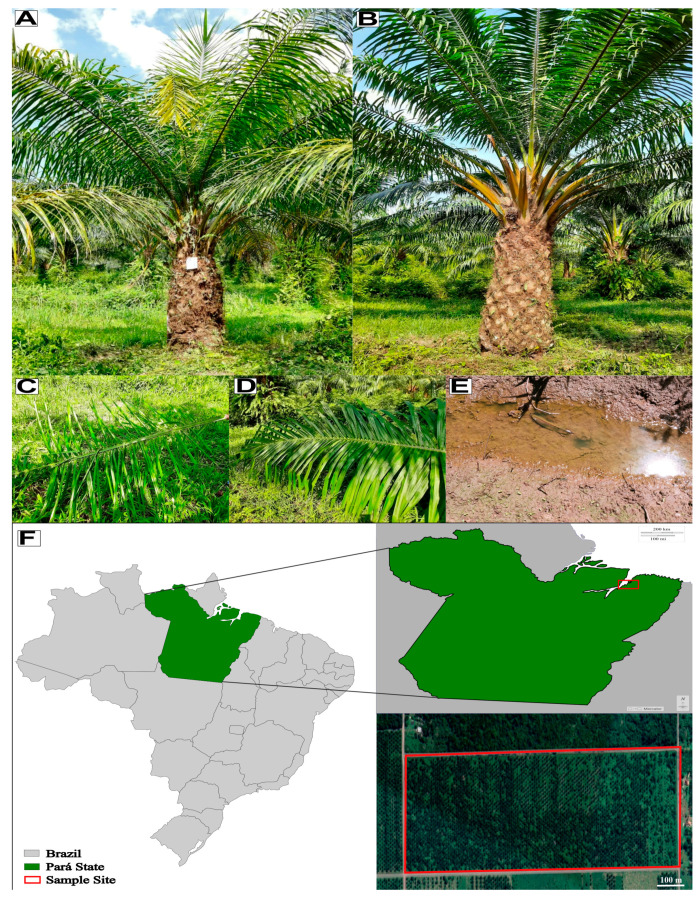
Overview of the sample site, soil condition and FY general phenotype; oil palm symptomatic (**A**) and asymptomatic (**B**) for FY; second leaf after spear leaf collected from a symptomatic (**C**) and asymptomatic (**D**) oil palm individual; waterlogged soil in the sample site from Dry period (**E**); sample site in Santa Bárbara do Pará, Pará, Brazil (**F**).

**Table 1 ijms-24-12918-t001:** List of metabolites identified in the leaves of oil palm affected by fatal yellowing in the dry period, after submitting the differentially expressed (DE) peaks to the pathway topology analysis module in MetaboAnalyst 5.0. FDR: False Discovery Rate; FC: Fold Change.

Query Mass	Matched Compound	Matched Form	Mass Diff	Compound Name	Log_2_(FC)	FDR	Profile
792.12440	C00024	M-H_2_O+H[1+]	1.95 × 10^−3^	Acetyl-CoA	−5.37	0.0004	Down
742.22112	C03541	M+K[1+]	2.18 × 10^−3^	THF-polyglutamate	−5.17	0.0012	Down
293.21349	C06427	M-H+O[-]	1.28 × 10^−3^	alpha-Linolenic acid	−1.11	0.0026	Down
312.16523	C16448	M-C_3_H_4_O_2_+H[1+]	1.43 × 10^−3^	Dihydrozeatin-O-glucoside	1.93	0.0111	Up
836.28348	C05275	M-HCOOK+H[1+]	2.18 × 10^−3^	trans-Dec-2-enoyl-CoA	−2.53	0.0220	Down
409.38261	C01054	M-H_2_O+H[1+]	2.33 × 10^−4^	(S)-2,3-Epoxysqualene	−4.13	0.0305	Down
425.37851	C22116	M-HCOOH+H[1+]	6.34 × 10^−4^	3beta-Hydroxy-4beta	−2.62	0.0305	Down
309.20812	C04785	M-H[-]	9.86 × 10^−4^	(9Z,11E,15Z)-(13S)-Hydroperoxyoctadeca-9,11,15-trienoate	−1.33	0.0346	Down
361.20077	C18016	M+HCOO[-]	7.15 × 10^−4^	3beta-Hydroxy-9beta-pimara-7,15-diene-19,6beta-olide	−1.20	0.0346	Down
426.38263	C22121	M(C13)+H[1+]	1.45 × 10^−3^	Cycloeucalenone	−2.26	0.0359	Down
407.36819	C03313	M-HCOOH+H[1+]	8.83 × 10^−4^	Phylloquinol	−1.90	0.0372	Down
87.00852	C00258	M-H_2_O-H[-]	2.12 × 10^−4^	D-Glycerate	0.75	0.0384	Up
446.16191	C00101	M(Cl37)-H[-]	1.00 × 10^−3^	Tetrahydrofolate	−0.64	0.0398	Down
129.01926	C06032	M-H_2_O-H[-]	3.35 × 10^−5^	D-erythro-3-Methylmalate	1.00	0.0401	Up
173.00911	C00311	M-H_2_O-H[-]	1.61 × 10^−5^	Isocitrate	1.00	0.0447	Up

**Table 2 ijms-24-12918-t002:** List of the pathways most affected in symptomatic plants, or in both symptomatic and asymptomatic plant at once, obtained via Multi-Omics Integration (MOI) using Omics Fusion, and with 10 or more enzymes and metabolites common to both phenotypes.

Pathway	Pathway ID	Common (Symptomatic and Asymptomatic)			Only in Symptomatic		
		Enzymes & Metabolites	Enzymes	Metabolites	Enzymes & Metabolites	Enzymes	Metabolites
Purine metabolism	230	32	15	17	9	4	5
Porphyrin and chlorophyll metabolism	860	29	10	19	4	3	1
Phenylpropanoid biosynthesis	940	20	4	16	4	2	2
Starch and sucrose metabolism	500	19	17	2	5	4	1
Glycolysis/Gluconeogenesis	10	17	14	3	12	9	3
Carbon fixation pathways in prokaryotes	720	17	5	12	8	2	6
Cysteine and methionine metabolism	270	16	8	8	10	7	3
Ubiquinone and other terpenoid-quinone biosynthesis	130	16	2	14	5	1	4
Pentose phosphate pathway	30	15	9	6	8	5	3
Aminoacyl-tRNA biosynthesis	970	14	12	2	3	3	0
Methane metabolism	680	14	8	6	10	7	3
Glyoxylate and dicarboxylate metabolism	630	14	7	7	5	3	2
Pyruvate metabolism	620	13	8	5	7	5	2
Glycerophospholipid metabolism	564	12	10	2	5	4	1
Glutathione metabolism	480	12	7	5	7	5	2
Citrate cycle (TCA cycle)	20	12	7	5	4	3	1
Glycine, serine and threonine metabolism	260	12	7	5	4	3	1
Galactose metabolism	52	11	6	5	5	4	1
Pyrimidine metabolism	240	11	4	7	4	1	3
Carotenoid biosynthesis	906	11	0	11	6	0	6
Flavonoid biosynthesis	941	11	0	11	4	0	4
Amino sugar and nucleotide sugar metabolism	520	10	9	1	5	4	1
Carbon fixation in photosynthetic organisms	710	10	7	3	7	5	2
Sulfur metabolism	920	10	6	4	4	1	3
Terpenoid backbone biosynthesis	900	10	5	5	4	2	2
Steroid biosynthesis	100	10	1	9	8	1	7
Biosynthesis of various secondary metabolites—part 2	998	10	0	10	2	0	2

**Table 3 ijms-24-12918-t003:** List of genes/proteins integrated in the purine metabolism (map00230), the top most affected pathway, and their behavior in FY asymptomatic and symptomatic oil palm plants, obtained via Multi-Omics Integration (MOI) using Omics Fusion.

Protein ID	UniProt Accession	EC Number	FC Symptomatic	Profile Symptomatic	FC Asymptomatic	Profile Asymptomatic
XP_010912022.1	A0A6I9QPT3	1.17.4.1	−4.0	DOWN	−2.3	DOWN
XP_010938967.1	A0A6I9S8I9	2.7.1.25	−2.5	DOWN	−3.0	DOWN
XP_010911123.2	A0A6I9QKC5	2.7.1.40	−5.9	DOWN	−3.6	DOWN
XP_010930617.1	A0A6I9RQ67	2.7.1.40	−1.8	DOWN	−2.8	DOWN
XP_010919863.2	A0A6I9R3I3	2.7.1.40	−2.9	DOWN	−2.2	DOWN
XP_010924524.1	A0A6I9RE71	2.7.4.6	−2.5	DOWN	−2.7	DOWN
XP_010937073.1	A0A6I9S4K9	2.7.4.8	−4.2	DOWN	−3.3	DOWN
XP_010910297.1	A0A6I9QJ47	2.7.6.5	−4.0	DOWN	−3.6	DOWN
XP_010933384.1	A0A6I9RX86	2.7.6.5	−2.6	DOWN	−2.9	DOWN
XP_010932410.1	A0A6I9RU27	2.7.6.5	−2.8	DOWN	−2.7	DOWN
XP_010921622.1	A0A6I9R798	2.7.6.5	−4.9	DOWN	−5.4	DOWN
XP_029119510.1	A0A8N4F2W4	2.7.7.4	−3.6	DOWN	−4.7	DOWN
XP_010932834.1	A0A6I9RV40	2.7.7.4	−2.2	DOWN	−2.4	DOWN
XP_010920819.1	A0A6I9R728	3.5.4.6	−3.3	DOWN	−3.7	DOWN
XP_010937877.2	A0A6I9S697	3.5.4.6	−4.6	DOWN	−2.2	DOWN
XP_029116569.1	A0A8N4EWM4	5.4.2.2	−2.0	DOWN	−3.0	DOWN
XP_010934074.1	A0A6I9RXY5	5.4.2.2	−1.9	DOWN	−2.1	DOWN
XP_010911922.1	A0A6I9QMW6	2.4.2.7	1.8	UP	−1.3	NDE
XP_010920467.1	A0A6I9R4T4	2.7.1.20	1.9	UP	1.2	NDE
XP_029117373.1	A0A8N4ID85	2.7.4.6	1.5	UP	−1.2	NDE
XP_010933513.1	A0A6I9RXJ3	2.7.1.40	−6.4	DOWN	No	No
XP_010905734.1	A0A6I9QAR4	1.7.3.3	2.1	UP	1.6	UP
XP_010907802.1	A0A6I9QFG8	1.7.3.3	3.2	UP	2.8	UP
XP_010941354.1	A0A6I9SCA5	2.7.4.3	1.7	UP	1.7	UP
XP_010908713.1	A0A6I9QHG6	2.7.4.3	1.8	UP	1.9	UP
XP_010935173.1	A0A6I9RZH1	2.7.4.3	4.6	UP	3.9	UP
XP_010919758.1	A0A6I9R9M7	2.7.4.6	2.4	UP	1.5	UP
XP_010933580.1	A0A6I9RXP4	2.7.4.6	3.6	UP	2.6	UP
XP_010943858.1	A0A6I9SHJ3	2.7.6.5	3.9	UP	2.8	UP
XP_010914531.2	A0A6I9QT08	6.3.3.1	1.7	UP	1.8	UP
XP_010910143.1	A0A6I9QKR6	6.3.4.13	2.2	UP	1.9	UP

**Table 4 ijms-24-12918-t004:** List of metabolites integrated in the purine metabolism (map00230), the top most affected pathway, and their behavior in FY asymptomatic and symptomatic oil palm plants, obtained via Multi-Omics Integration (MOI) using Omics Fusion.

KEGG ID	Compound	Matched Form Symptomatic	Fold Change Asymptomatic	Profile Asymptomatic	Fold Change Symptomatic	Profile Symptomatic
C00104	IDP	M-HCOOK+H[1+]	0.07	DOWN	87.59	UP
C06197	P1,P3-Bis(5′-adenosyl) triphosphate	M+NaCl[1+]	0.16	DOWN	0.05	DOWN
C00212	Adenosine	M+Cl[-]	0.30	DOWN	4.97	UP
C00387	Guanosine	M+Na[1+]	0.52	DOWN	2.54	UP
C04640	2-(Formamido)-N1-(5′-phosphoribosyl) acetamidine	M+HCOONa[1+]	0.27	DOWN	0.20	DOWN
C12248	5-Hydroxy-2-oxo-4-ureido-2,5-dihydro 1H-imidazole-5-carboxylate	M[1+]	0.42	DOWN	2.88	UP
C00242	Guanine	M+Na[1+]	0.12	DOWN	9.74	UP
C00224	Adenylyl sulfate	M-NH_3_+H[1+]	0.19	DOWN	2.74	UP
C00655	Xanthosine 5′-phosphate	M+NaCl[1+]	0.37	DOWN	2.98	UP
C04823	1-(5′-Phosphoribosyl)-5-amino-4 (N-succinocarboxamide)-imidazole	M-HCOOH+H[1+]	0.20	DOWN	7.89	UP
C00301	ADP-ribose	M-H_2_O-H[-]	No	No	0.22	DOWN
C00385	Xanthine	M+Na-2H[-]	No	No	7.90	UP
C00206	dADP	M+3H[3+]	No	No	14.49	UP
C02091	(S)-Ureidoglycine	M[1+]	No	No	2.51	UP
C00059	Sulfate	M(S34)-H[-]	No	No	2.27	UP
C04677	1-(5′-Phosphoribosyl)-5-amino-4 imidazolecarboxamide	M-H[-]	10.95	UP	66.60	UP
C00130	IMP	M-H_4_O_2_+H[1+]	2.93	UP	112.58	UP

## Data Availability

The datasets used and/or analyzed in the current study are available from the corresponding author on reasonable request.

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
