# Peer review of "Molecular Interplay between Non-Host Resistance, Pathogens and Basal Immunity as a Background for Fatal Yellowing in Oil Palm (Elaeis guineensis Jacq.) Plants"

_ijms, 2023, doi:10.3390/ijms241612918_

Round 1

Reviewer 1 Report

This is very original and actual work. Though it is focused on particular problem of oil palm disease, it presents good example of complex study of the molecular mechanism governing plant development.

I suggest make shorter title, update the keywords list. Update the Abstract.

Currently, the title is too long. Try make it a little shorter.

My suggestion -

Molecular interplay between non-host resistance, pathogens and the basal immunity as background for Fatal Yellowing in Oil Palm (Elaeis guineensis Jacq.) plants

It is on the authors’ discretion.

This is ‘Molecular Sciences’ journal – it is good to have in keywords ‘molecular mechanisms’ or like that.

 Line 23: “all of Koch's postulates” – please comment, name shortly the postulates.

 Line 26: (SOA) and (<OI) – remove these abbreviations from the Abstract. It used once anyway. MOI abbreviation used once more. But please avoid any abbreviation in the Abstract.

Please update the keywords -add ‘molecular plant biology’, pathogen, palm oil, Fatal Yellowing..

Line 56: ‘Legal Amazon Area’ – please comment on this term. It is not understandable for the journal readers.

Use simple English, not so pathetic, not need mention Brazil many times. The journal readers will read about molecular mechanisms of plant pathogen, that might be similar for other plants, but not about the economy.

Please rephrase, or remove “…contrary to expectations, Brazil has not significantly increased its palm oil harvest area and has not become a world player in the palm oil industry” –

 Just write like that “… the problem of FY study is very actual, especially in Brazil.’

Line 66: ‘a complex and multi-factor scenario in Brazil.’ – please comment about other factors, add references about this scenario.

 Line 76: ‘third postulate’ – please add details about this postulate.

 Line 84: ‘Boron (B) and Cu’ – use word ‘Copper’ too. Or use only chemical elements signs, not mix in the same sentence.

 Line 90: ‘Recent studies…’ – add references to this sentence.

 Line 95: ‘Pudricion del Cogollo’ – this is not understandable, please extend the phrase. Not need use abbreviation ‘PC’.

 Line 125: ‘In the soil, Carbon, Chlorine, Na, P, and Zn..’ -  mixture of elements names and signs in one phrase, use standard names…

Line 132: ‘Ca/CEC and K/CEC’ – please comment what is CEC here?

 Line 134: ‘Mn, Zn, Na, Al, pH, CEC, OM, silt, and clay’ – please ass word ‘level’ or ‘concentration’, comment about CEC and OM – it is not clear now, mixture of many parameters (and in what scale?)

 Figure 3 has too many small panels. Please make it larger.

Place  only 3-4 panels (histograms) in row (up to the page width). So, the signs will be larger.

Line 156: ‘1,924, 576, 2,469,  and 272 peaks…’ – please extend the phrase. What these number mean? Too many chemical components or what? Look like redundant information.

 Line 163: ‘mummichog and GSEA pathways’ – add the references to these tools.

 Table 1 has grey rows (after Propenoyl-CoA)

It seems these rows are statistically significant. Then rearrange it, start from the top, mark by * sign, comment about in the Table Note.

Line 225: ‘FC ≠ 1’ – pleas rewrite. Maybe |FC|>1.5 or like that. Not use sign ≠

Table 2 is too large. Try fit to 1 page. If the table is so large, then move it to Supplementary materials, and show only top rows.

Line 292: ‘In 2024, Brazil will celebrate 50 years of the first FY appearance in the country. Of course, there is nothing to celebrate; on the contrary.’ – change this phrase. Use less pathetic language.

Line 341: ‘(Denpasa's staff - Personal communication).  -  comment what is Denpasa, give here the name. Please format is as proper reference.

 Table 3. List of metabolites and genes… -

This is interesting table, but too large. Please separate it to two tables, separately for genes and separately for metabolites. Then it will be more compact and readable, each table fits one page size..

Figure 8 is good. It could be first figure to show the idea of the experiment. I would recommend add arrow or some mark on first panel to show how Fatal Yellowing looks like. In the photo it is similar in color to green, not very yellow.

Line 576: ‘The one-way analysis of variance (ANOVA) investigated..’ = phrase is not correct, the analysis can’ investigate. Then write like ‘We investigated …using the one-way analysis of variance (ANOVA)’

 Line 590: ‘PRINCOMP Procedure and Anova´s using the GLM (General Linear Model) Procedure from the statistica l software SAS/STAT® …’ = please rephrase. Write first about the statical methods, then mention the tool used. Not just mention the program/routine from the tool.

Line 683: ‘The data underwent enrichment through several databases (EMBL, KEGG, NCBI, and UniProt)…’ – [lease cite all these tools – EMBL, NCI, at least by URL / web-link. Indicate version of the database used.

 Lines 729 and 733 : ‘pave the way’ – the phrase is repeated, change it.

 Reference 3 - [Online Resource] – cite with the Access date.

Please avoid ambitious language forms

Author Response

RESPONSE TO REVIEWER 01

Comments and Suggestions for Authors

This is very original and actual work. Though it is focused on particular problem of oil palm disease, it presents good example of complex study of the molecular mechanism governing plant development.

I suggest make shorter title, update the keywords list. Update the Abstract.

Currently, the title is too long. Try make it a little shorter.

My suggestion - Molecular interplay between non-host resistance, pathogens and the basal immunity as background for Fatal Yellowing in Oil Palm (Elaeis guineensis Jacq.) plants

It is on the authors’ discretion.

RESPONSE: Thank you very much for your suggestion. We decided to accept it, and changed the title of the manuscript using your suggestion as it was. Please, see the new version of the manuscript.

This is ‘Molecular Sciences’ journal – it is good to have in keywords ‘molecular mechanisms’ or like that.

RESPONSE: Thank you very much for your suggestion. We decided to accept it. Please, see the new version of the manuscript.

Line 23: “all of Koch's postulates” – please comment, name shortly the postulates.

RESPONSE: Thank you very much for your suggestion. We decided to accept it. Please, see the new version of the manuscript.

Line 26: (SOA) and (<OI) – remove these abbreviations from the Abstract. It used once anyway. MOI abbreviation used once more. But please avoid any abbreviation in the Abstract.

RESPONSE: Thank you very much for your suggestion. We decided to accept it. Please, see the new version of the manuscript.

Please update the keywords -add ‘molecular plant biology’, pathogen, palm oil, Fatal Yellowing..

RESPONSE: Thank you very much for your suggestion. We decided to accept it. Please, see the new version of the manuscript.

Line 56: ‘Legal Amazon Area’ – please comment on this term. It is not understandable for the journal readers.

RESPONSE: Thank you very much for your question. The answer is: "The Legal Amazon, known as "Amazônia Legal" in Portuguese, is an area of more than five million square kilometers comprising the Brazilian states of Acre, Amapá, Amazonas, Maranhão, Mato Grosso, Pará, Rondônia, Roraima, and Tocantins. A region that occupies almost half of all Brazilian territory, covering 9 states and an area larger than the Amazon biome itself". A suggestion for further information regarding the area in Brazil can be found in: Müller-Hansen, F., Cardoso, M. F., Dalla-Nora, E. L., Donges, J. F., Heitzig, J., Kurths, J., and Thonicke, K.: A matrix clustering method to explore patterns of land-cover transitions in satellite-derived maps of the Brazilian Amazon, Nonlin. Processes Geophys., 24, 113–123, https://doi.org/10.5194/npg-24-113-2017, 2017.

Use simple English, not so pathetic, not need mention Brazil many times. The journal readers will read about molecular mechanisms of plant pathogen, that might be similar for other plants, but not about the economy.

RESPONSE: Thank you very much for your suggestion. We decided to accept it. Please, see the new version of the manuscript.

Please rephrase, or remove “…contrary to expectations, Brazil has not significantly increased its palm oil harvest area and has not become a world player in the palm oil industry” – Just write like that “… the problem of FY study is very actual, especially in Brazil.’

RESPONSE: Thank you very much for your suggestion. We decided to accept it. Please, see the new version of the manuscript.

Line 66: ‘a complex and multi-factor scenario in Brazil.’ – please comment about other factors, add references about this scenario.

RESPONSE: Thank you very much for your comment. We made some changes in the text, but we decided not to comment about other factors. Some of those factors are land disputes, logistics, long distance from the main markets in the country, and so on. In terms of cultivation of the oil palm plant, AF is indubitably the most important problem affecting oil palm in Brazil.

Line 76: ‘third postulate’ – please add details about this postulate.

RESPONSE: Thank you very much for your suggestion. We decided to accept it. Please, see the new version of the manuscript.

Line 84: ‘Boron (B) and Cu’ – use word ‘Copper’ too. Or use only chemical elements signs, not mix in the same sentence.

RESPONSE: Thank you very much for your suggestion. We decided to accept it. Please, see the new version of the manuscript.

Line 90: ‘Recent studies…’ – add references to this sentence.

RESPONSE: Thank you very much for your suggestion. We decided to accept it. Please, see the new version of the manuscript.

Line 95: ‘Pudricion del Cogollo’ – this is not understandable, please extend the phrase. Not need use abbreviation ‘PC’.

RESPONSE: Thank you very much for your suggestion. We decided to accept it. Please, see the new version of the manuscript.

Line 125: ‘In the soil, Carbon, Chlorine, Na, P, and Zn..’ - mixture of elements names and signs in one phrase, use standard names…

RESPONSE: Thank you very much for your suggestion. We decided to accept it. Please, see the new version of the manuscript.

Line 132: ‘Ca/CEC and K/CEC’ – please comment what is CEC here?

Line 134: ‘Mn, Zn, Na, Al, pH, CEC, OM, silt, and clay’ – please ass word ‘level’ or ‘concentration’, comment about CEC and OM – it is not clear now, mixture of many parameters (and in what scale?)

RESPONSE: Thank you very much for your comment. The detailed information is present in Figure 2 and Supplementary Table 2. So, instead of changing the text as it is now, we decided just to point out that the details can be found in that figure and that table. Please, see the new version of the manuscript.

Figure 3 has too many small panels. Please make it larger. Place only 3-4 panels (histograms) in row (up to the page width). So, the signs will be larger.

RESPONSE: Thank you very much for your suggestion. We decided to partially accept it. Please, see the new version of the manuscript.

Line 156: ‘1,924, 576, 2,469, and 272 peaks…’ – please extend the phrase. What these number mean? Too many chemical components or what? Look like redundant information.

RESPONSE: Thank you very much for your question. As a consequence, we did changes in the text. Please, see the new version of the manuscript. That is not a redundant information, as it shows with how many peaks we start the analysis, and it is a reference to measure the efficiency of untargeted metabolomics as a tool to identify metabolites nowadays; which is still too low.

Line 163: ‘mummichog and GSEA pathways’ – add the references to these tools.

RESPONSE: Thank you very much for your suggestion. We add a comment pointing out that additional information on those tools are in reference [59] - Pang, Z., Chong, J., Zhou, G., de Lima Morais, D. A., Chang, L., Barrette, M., Gauthier, C., Jacques, P. É., Li, S., & Xia, J. (2021). MetaboAnalyst 5.0: narrowing the gap between raw spectra and functional insights. Nucleic acids research, 49(W1), W388–W396. https://doi.org/10.1093/nar/gkab382. Please, see the new version of the manuscript.

Table 1 has grey rows (after Propenoyl-CoA) - It seems these rows are statistically significant. Then rearrange it, start from the top, mark by * sign, comment about in the Table Note.

RESPONSE: Thank you very much for your comment. Actually, we left those metabolites with FDR were between 0.5 and 0.6, which had also been used to run the pathway topology analysis module in MetaboAnalyst 5.0. But only those 0.5 are considered statistically significant. To avoid misunderstanding, we decide to remove them from the Table 1. Please, see the new version of the manuscript.

Line 225: ‘FC ≠ 1’ – please rewrite. Maybe |FC|>1.5 or like that. Not use sign ≠

RESPONSE: Thank you very much for your comment. Actually, it is not a case of absolute value. What we intended to represent is that once a transcript/gene/protein is considered statistically differentially expressed with a FDR 0.05, we apply the criteria FC different of 1.0, not FC

Table 2 is too large. Try fit to 1 page. If the table is so large, then move it to Supplementary materials, and show only top rows.

RESPONSE: Thank you very much for your comment. We tried to reduce it to fit one page, but did not work well. So, we decide to maintain only the data about the pathways with 10 or more enzymes and metabolites; which were 27 pathways. This new version of the table has the most important pathways, and fits to one page. Please, see the new version of the manuscript.

Line 292: ‘In 2024, Brazil will celebrate 50 years of the first FY appearance in the country. Of course, there is nothing to celebrate; on the contrary.’ – change this phrase. Use less pathetic language.

RESPONSE: Thank you very much for your comment. We decided to accept it. Please, see the new version of the manuscript.

Line 341: ‘(Denpasa's staff - Personal communication). - comment what is Denpasa, give here the name. Please format is as proper reference.

RESPONSE: Thank you very much for your comment. We had already a more complete description of Denpasa in the Materials & Methods part, but we did follow you suggestion and made a better description in Line 341 as well. Please, see the new version of the manuscript.

Table 3. List of metabolites and genes… -

This is interesting table, but too large. Please separate it to two tables, separately for genes and separately for metabolites. Then it will be more compact and readable, each table fits one page size..

RESPONSE: Thank you very much for your suggestion. We decided to accept it. Please, see the new version of the manuscript.

Figure 8 is good. It could be first figure to show the idea of the experiment. I would recommend add arrow or some mark on first panel to show how Fatal Yellowing looks like. In the photo it is similar in color to green, not very yellow.

RESPONSE: Thank you very much for your suggestion. We decided to keep it as it is now. We agree that it could be Figure 1 (and it was in the first version of the manuscript), but it was cited in Materials & Methods (in IJMS it comes after the discussion part, then we had to change and it became Figure 8).

Line 576: ‘The one-way analysis of variance (ANOVA) investigated..’ = phrase is not correct, the analysis can’ investigate. Then write like ‘We investigated …using the one-way analysis of variance (ANOVA)’

RESPONSE: Thank you very much for your suggestion. We decided to accept it. Please, see the new version of the manuscript.

Line 590: ‘PRINCOMP Procedure and Anova´s using the GLM (General Linear Model) Procedure from the statistica l software SAS/STAT® …’ = please rephrase. Write first about the statical methods, then mention the tool used. Not just mention the program/routine from the tool.

RESPONSE: Thank you very much for your suggestion. We decided to accept it. Please, see the new version of the manuscript.

Line 683: ‘The data underwent enrichment through several databases (EMBL, KEGG, NCBI, and UniProt)…’ – [lease cite all these tools – EMBL, NCI, at least by URL / web-link. Indicate version of the database used.

RESPONSE: Thank you very much for your suggestion. We decided to accept it. Please, see the new version of the manuscript.

Lines 729 and 733 : ‘pave the way’ – the phrase is repeated, change it.

RESPONSE: Thank you very much for your suggestion. We decided to accept it. Please, see the new version of the manuscript.

Reference 3 - [Online Resource] – cite with the Access date.

RESPONSE: Thank you very much for your suggestion. We decided to accept it. Please, see the new version of the manuscript.

Comments on the Quality of English Language

Please avoid ambitious language forms

RESPONSE: Thank you very much for your suggestion. We decided to accept it. Please, see the new version of the manuscript.

Reviewer 2 Report

The manuscript entitled “Breaks in the non-host resistance to non-adapted pathogens and the basal immunity to adapted pathogens, triggered by waterlogged soil, might be in the onset of Fatal Yellowing in Oil Palm plants, a disease of unknown etiology” by Bittencourt et al, studied a bud rod disorder of unknown etiology for Fatal Yellowing (FY) disease. 

They collected the leaves of oil palms between symptomatic plants and the asymptomatic ones from two distinct seasons (dry and rainy), and employed a comprehensive, large-scale, single- (SOA) and multi-omics integration (MOI) analysis to compare the transcriptomics and metabolomics data. The preliminary results showed the possible molecular mechanisms of oxygen deficiency as the initial cause of FY. This study might be useful to develop biomarkers for selecting oil palm plants resistant to this disease and to pave the way to employ strategies to keep the safety barriers up and strong. 

Overall, this design of the study and sample collection is sound, which provides comparisons between four scenarios, and many valuable data for multi-omics analysis. Conclusions are appropriate, and supported by the data. Statistical analysis is provided within the manuscript. I recommend accepting it in the present form. 

Author Response

RESPONSE TO REVIEWER 02

Comments and Suggestions for Authors: The manuscript entitled “Breaks in the non-host resistance to non-adapted pathogens and the basal immunity to adapted pathogens, triggered by waterlogged soil, might be in the onset of Fatal Yellowing in Oil Palm plants, a disease of unknown etiology” by Bittencourt et al, studied a bud rod disorder of unknown etiology for Fatal Yellowing (FY) disease.

They collected the leaves of oil palms between symptomatic plants and the asymptomatic ones from two distinct seasons (dry and rainy), and employed a comprehensive, large-scale, single- (SOA) and multi-omics integration (MOI) analysis to compare the transcriptomics and metabolomics data. The preliminary results showed the possible molecular mechanisms of oxygen deficiency as the initial cause of FY. This study might be useful to develop biomarkers for selecting oil palm plants resistant to this disease and to pave the way to employ strategies to keep the safety barriers up and strong.

Overall, this design of the study and sample collection is sound, which provides comparisons between four scenarios, and many valuable data for multi-omics analysis. Conclusions are appropriate, and supported by the data. Statistical analysis is provided within the manuscript. I recommend accepting it in the present form.

RESPONSE: Thank you very much for your comments. Although you did not suggest any modification in the text, we generated and submitted a revised version based on the comments and suggestions of other reviewers. Please, see the new version of the manuscript. Your opinion on this new version is important.